# *GMDS* Intragenic Deletions Associate with Congenital Heart Disease including Ebstein Anomaly

**Shirley M. Lo-A-Njoe** [1,2,†], **Eline A. Verberne** [3,†], **Lars T. van der Veken** [4], **Eric Arends** [1], **J. Peter van Tintelen** [4], **Alex V. Postma** [3,5] **and Mieke M. van Haelst** [3,*]

1   Department of Pediatrics, Dr. Horacio E. Oduber Hospital, Oranjestad, Aruba
2   Department of Pediatrics, Curaçao Medical Center, Willemstad, Curacao
3   Department of Human Genetics, Amsterdam UMC, 1100 DD Amsterdam, The Netherlands
4   Department of Genetics, Division Laboratories, Pharmacy and Biomedical Genetics, University Medical Center Utrecht, 3584 CX Utrecht, The Netherlands
5   Department of Medical Biology, Amsterdam UMC, 1100 DD Amsterdam, The Netherlands
\*   Correspondence: m.vanhaelst@amsterdamumc.nl; Tel.: +31-651401743
†   These authors contributed equally to this work.

**Abstract:** Ebstein anomaly is a rare heterogeneous congenital heart defect (CHD) with a largely unknown etiology. We present a 6-year-old girl with Ebstein anomaly, atrial septum defect, hypoplastic right ventricle, and persistent left superior vena cava who has a de novo intragenic ~403 kb deletion of the GDP-mannose 4,6-dehydratase (*GMDS*) gene. *GMDS* is located on chromosome 6p25.3 and encodes the rate limiting enzyme in GDP-fucose synthesis, which is used to fucosylate many proteins, including Notch1, which plays a critical role during mammalian cardiac development. The *GMDS* locus has sporadically been associated with Ebstein anomaly (large deletion) and tetralogy of Fallot (small deletion). Given its function and the association with CHD, we hypothesized that loss-of-function of, or alterations in, *GMDS* could play a role in the development of Ebstein anomaly. We collected a further 134 cases with Ebstein anomaly and screened them for genomic aberrations of the *GMDS* locus. No additional *GMDS* genomic aberrations were identified. In conclusion, we describe a de novo intragenic *GMDS* deletion associated with Ebstein anomaly. Together with previous reports, this second case suggests that *GMDS* deletions could be a rare cause for congenital heart disease, in particular Ebstein anomaly.

**Keywords:** Ebstein anomaly; congenital heart defects; GDP-mannose 4,6-dehydratase; *GMDS*; 6p25.3 deletion

## 1. Introduction

Ebstein anomaly is a rare congenital heart defect in which the tricuspid valve is malformed and displaced downward into the right ventricle.

It accounts for less than 1% of all congenital heart diseases. Older studies reported a live birth prevalence of Ebstein anomaly of around 1:200,000 [1], but more recent population studies have reported a lower birth prevalence of around 1:20,000 [2–4]. This has been attributed to different factors, i.e., earlier and better diagnoses [5], better technology, and differences in ascertainment methods and classifications [5,6].

The clinical presentation of Ebstein anomaly varies widely and depends on the degree of anatomic abnormalities. Additional cardiac anomalies, such as patent foramen ovale, atrial septum defect, pulmonary stenosis or atresia, and ventricular septum defect, are often present in cases with Ebstein anomaly [7]. Approximately 20% of cases with Ebstein anomaly have extra-cardiac malformations suggesting that the cardiac defect could be part of a genetic syndrome [2]. The etiology of Ebstein anomaly is largely unknown. Epidemiologic studies have reported associations between Ebstein anomaly and environmental factors, such as peri-conceptional exposure to pesticides and varnishes as well as maternal

infections and maternal health conditions [2,3,7]. In addition, familial recurrence and iden-
tification of genetic variants segregating with this congenital disorder also suggest a genetic
component [8,9]. Although Ebstein anomaly has been sporadically described in genetic
syndromes, including Down, CHARGE, Noonan, and Cornelia de Lange syndrome [10–13],
a consistent association between Ebstein anomaly and a distinct genetic condition has not
yet been identified [10]. Pathogenic variants in three genes (MYH7, NKX2-5, and GATA4)
and different copy number variants have been described in cases with Ebstein anomaly
but only account for a small proportion of cases [6,13–15]. In 2016, Sicko et al. performed
a genome-wide investigation of copy number variants in a cohort of 47 cases with iso-
lated Ebstein anomaly [6]. They identified several (new) candidate copy number variants,
including one intragenic *GMDS* deletion of approximately 345 kb on chromosome 6p25.3.

Here, we report a second case of Ebstein anomaly associated with a de novo 403
kb intragenic *GMDS* deletion. Additionally, we analyzed a further 134 cases with iso-
lated Ebstein anomaly for abnormalities in the *GMDS* genomic locus, specifically copy
number variants.

## 2. Materials and Methods

### 2.1. Clinical Data

The patient was enrolled at Amsterdam UMC in Amsterdam, the Netherlands. The
parents provided written consent for research use of their data, and the study was con-
ducted in accordance with the Declaration of Helsinki. Clinical data were retrieved from the
patient's records. A total of 134 additional patients were evaluated from the Netherlands
and the patient's island of origin. We obtained genomic DNA from 129 cases with the main
clinical diagnosis of either Ebstein malformation, congenital pulmonary valve stenosis, or
tricuspid atresia. These cases were collected through the Dutch national biobank of adult
cases with congenital heart defects (CONCOR) [16]. Subsequently, we screened these cases
for copy number variants (CNVs) in *GMDS* using digital droplet PCR for three locations
across the whole *GMDS* locus, including the previously identified deletion. Using this
method, we could positively identify the deletion in the proband, validating the array-CGH
results. From the same Caribbean island, genomic DNA from five cases with Ebstein
anomaly was analyzed. Under Dutch law, assessment of the study protocol by our ethics
committee was not required since only genetic and clinical data collected during regular
patient care were used. This is specifically explained for (Dutch) research involving the
CONCOR registry and DNA-bank [16,17].

### 2.2. Array-CGH

Copy number profiling was performed on DNA isolated from peripheral blood using
180 K (Amadid #023363) Human Genome CGH Microarray slides from Agilent Technologies
(Version 5.1, Santa Clara, CA, USA) according to the manufacturer's protocols. The results
were classified with Cartagenia BENCH software 5.1. (Cartagenia, Leuven, Belgium).

### 2.3. Next-Generation Sequencing Targeted Panel

In addition, a next-generation sequencing (NGS) targeted structural heart disease
gene panel was performed, which consisted of 46 genes, including Noonan-syndrome
related genes (*BRAF*, *KRAS*, *LZTR1*, *MAP2K1*, *NRAS*, *PTPN11*, *RAF1*, *RASA2*, *RIT1*, *SOS1*,
and *SOS2*).

### 2.4. CNV Detection Using Droplet Digital PCR (ddPCR)

For ddPCR, the QX200 Droplet Digital PCR system (Bio-Rad, Hercules, CA, USA) was
used. The ddPCR reaction was performed in a 21 µL volume consisting of 10 µL 2× ddPCR
Supermix for Probes (No UDP) (Bio-Rad), 1 µL 20× target primers/probe mix (FAM), 1 µL
20× reference primers/probe mix, 1 µL (5U) Hind111 restriction enzyme (New England
Biolabs, Ipswich, MA, USA), and 1 µL DNA sample at 20 ng/µL. The mixture was loaded
into a DG8 cartridge (Bio-Rad) together with 70 µL of droplet generation oil (Bio-Rad) and

covered with a DG8 gasket. After processing the droplets in the droplet generator, the samples were transferred to a 96-well PCR plate. PCR amplification was carried out in a T11 Touch thermal cycler (Bio-Rad). The cycling protocol was as follows: 95 °C for 10 min, 40 cycles of 94 °C for 30 s and 60 °C for 1 min followed by an infinite 15-degree hold. After PCR, the plate was loaded on the QX200 droplet reader (Bio-Rad). Data were analyzed using QuantaSoft Analysis Pro software Version 1.2 (Bio-Rad).

## 3. Results

### 3.1. The Clinical Description of the Affected Patient Is Presented in This Section

The female index case was born to healthy Caribbean parents after an uneventful pregnancy of 37.5 weeks. She had a birth weight of 3200 g (50th centile), length of 51 cm (85th centile), and head circumference of 32.5 cm (10th centile). On the second day of life, she was admitted to the neonatology ward due to cyanosis with a heart rate of 125/min, respiratory rate of 40/min, and oxygen saturation of 84%. Echocardiography revealed an Ebstein anomaly with an atrial septal defect, hypoplastic right ventricle, and persistent left superior vena cava. Treatment with prostaglandin E2 and nasal continuous positive airway pressure was started. She was then referred to a tertiary pediatric heart center in Colombia for surgery.

The patient underwent a Starnes procedure, atrio-septostomy, closure of the pulmonary valve, and placement of a 3.5 mm Blalock–Taussig shunt. She experienced postoperative complications, including renal failure and paralysis of the diaphragm. At the age of one year, she had a follow-up surgery consisting of a bidirectional Glenn shunt. She suffered from bronchiolitis, pneumonia, and asthma and required a gastrostomy tube for feeding. At the age of 9 months, she was examined by a clinical geneticist. At that time, her height was 65 cm (2nd centile), weight was 6820 g (15th centile), and head circumference was 42 cm (15th centile). Dysmorphological examination showed hypertelorism, mid face hypoplasia, long philtrum, uplifted earlobes, and an anteriorly placed anus (Figure 1). An abdominal ultrasound was performed at the age of 10 months to exclude additional organ abnormalities.

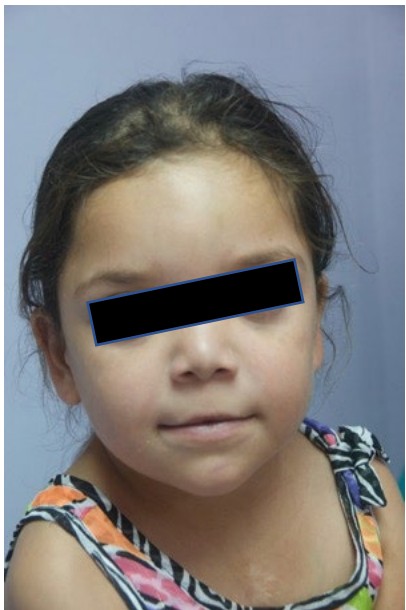

**Figure 1.** Photograph of the index patient at four years of age. Hypertelorism, long flat philtrum, and mid face hypoplasia can be recognized.

At the age of 4 years, she developed severe cyanosis caused by an aorta-pulmonary collateral, which was closed by intervention. Because of severe pulmonary hypertension, completion of Fontan circulation was not feasible. She had divergent eyes, a difference

in size of the globe, and a right nasolacrimal duct stenosis. She also had delayed motor and speech development and developed convulsions at the age of one year. EEG showed occipital epileptic activity for which anti-epileptic therapy was started.

Because of left-sided hypotonia at the age of 2 years, a CT of the brain was performed, which showed no abnormalities. This has resolved spontaneously.

### 3.2. Molecular and Cytogenetic Studies

An array-CGH was performed. The Agilent Bench Lab CNV Version 5.1 180 K array-CGH showed a de novo deletion on chromosome 6p25.3 of ~403 kb (Figure 2) and a maternally inherited duplication in Xq27.2 of ~316 kb (arr [GRCh37] 6p25.3(1727928_2131157) x1 dn, Xq27.2(140693307_141009050) x3 mat, in accordance with the ISCN 2020 nomenclature [18]. No (likely) pathogenic variants were identified in the NGS gene panel.

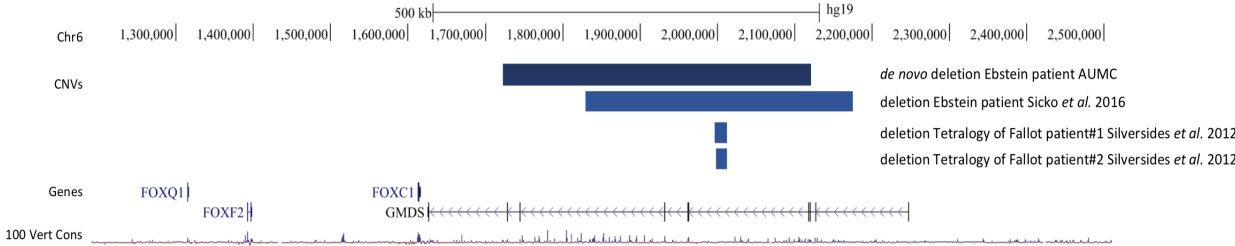

**Figure 2.** *GMDS* genomic locus (hg19), demonstrating the location of surrounding genes and the location of both the de novo deletion from the index patient and the previously published *GMDS* deletion by Sicko et al. [6] and Silversides et al. [19].

### 3.3. Genetic Investigations in Large Ebstein Anomaly Cohort

Genetic investigations in 129 Ebstein cases from the CONCOR data revealed no additional deletions or duplications.

### 3.4. Genetic Investigations in Caribbean Ebstein Cohort

In the five Caribbean cases with Ebstein anomaly, SNP arrays were performed, but none of them showed a 6p25.3 deletion. One patient had a 16q23.1 deletion, but unfortunately, her parents refused testing, and the other patients all had normal SNP array results.

## 4. Discussion

We report a patient with Ebstein anomaly, a rare, heterogeneous congenital heart defect with largely unknown etiology. Although Ebstein anomaly has been associated with pathogenic variants in a handful of genes and various copy number variants, these account for only a small proportion of cases [4,11–13]. We demonstrate here that the index patient has a large, intragenic, and de novo deletion in *GMDS* that is likely associated with the patient's phenotype. *GMDS* encodes a short-chain mannose dehydrogenase protein (GDP-mannose 4,6-dehydratase) that catalyzes the first step in a molecular process called fucosylation, which has a function in many processes important in development [20]. *GMDS* is a key player in fucosylation as it is the rate-limiting enzyme [20]. Although no pathogenic human single nucleotide variations have been reported for *GMDS*, a case with Ebstein anomaly and a comparable large intragenic *GMDS* deletion has been recently reported [6]. The inheritance of that deletion was not reported. Both these large intragenic deletions are predicted to lead to loss-of-function of *GMDS* as they remove a large part of the *GMDS* coding region. In addition, two patients with Tetralogy of Fallot (TOF), a rare congenital heart disease, have been described to harbor a smaller deletion (of unknown inheritance) in the same region as the two Ebstein patients [19] (see Figure 2). These TOF-associated deletions, however, do not delete any *GMDS* coding parts, making it difficult to predict their effect. Nonetheless, based on our results coupled with the cases described above, we hypothesize that *GMDS*, or the *GMDS* locus, is important for normal

(cardiac) development. Animal experiments corroborate this as *GMDS* knockout mice have an abnormal cranium and snout morphology [21], while Song et al. demonstrated that a *GMDS* missense variant in zebrafish results in disturbed neural development and reduced *Notch* signaling (see below) [22]. Together, this indicates that *GMDS* or its products have a role in normal development. One possible mechanism by which *GMDS* might be involved in development comes from its role as a major player in protein glycosylation (specifically fucosylating). It is known that protein glycosylation regulates various functions and physicochemical properties and that it plays a critical role in maintaining homeostasis. However, alterations of glycosylation are associated with the development and aggravation of many diseases [23]. This raises the possibility that altered glycosylation could lead to congenital heart disease as observed in our patient and the other three reported intragenic *GMDS* deletion patients. A possible link between congenital heart disease and altered glycosylation might be that a group of key proteins in (cardiac) development, the *Notch* proteins, are actually fucosylated [24]. Indeed, reduced fucose levels result in decreased *Notch1* activation [23,24], a key player in cardiac development [25,26].

The importance of *Notch1* in cardiac development is underscored by the fact that it is one of the most frequently mutated genes in patients with congenital heart disease, and that *Notch1* KO mice demonstrate a plethora of congenital heart disease phenotypes [26]. One hypothesis could therefore be that *GMDS* haploinsufficiency leads to decreased GDP-fucose production, which in turn affects fucosylation of *Notch1*, thereby reducing its activity, resulting in cardiac anomalies. Alternatively, it is also possible that the intragenic deletions not only impact *GMDS* but also its nearest neighbor *FOXC1*, a forkhead transcription factor [27] located just 10 kb from the last *GMDS* exon. *FOXC1* is an essential component of mesodermal [27] and neural crest development [27], both important for heart development [28,29]. While *FOXC1* is not affected by any of the deletions directly, it is possible that disruption of conserved enhancer sequences within the *GMDS* locus also affects the regulation of (spatial-temporal) expression of *FOXC1*. *FOXC1*-knockout mice present with a variety of phenotypes, including congenital heart disease, and demonstrate abnormal valve development, which is part of the Ebstein anomaly, suggesting that altered expression of *FOXC1* may cause developmental problems of the heart [29]. An additional possibility is that *GMDS* locus deletions have an impact on multiple neighboring genes and thereby cause the phenotype. Regardless, further investigation into the mechanism of how intragenic *GMDS* deletions can lead to disease is needed.

Given the link between *GMDS* and congenital heart disease, we collected a cohort of 134 Ebstein and Ebstein like cases, in which we investigated the *GMDS* locus for additional aberrations. No deletions or duplication were identified in this cohort, suggesting that *GMDS* intragenic aberrations are a relatively rare cause of Ebstein anomaly.

In addition to Ebstein anomaly, our index case also presented with seizures and dysmorphic facial features. This is in contrast to the other large *GMDS* intragenic deletion case, where the Ebstein anomaly was reported as an isolated condition [11]. However, as no additional clinical details were reported, it is possible that minor birth defects, such as dysmorphic features, and/or developmental delay were present in that patient. Alternatively, the difference in phenotype could also be caused by differences in genetic background and/or the difference in size and location of the *GMDS* intragenic deletions. With regard to the two cases with smaller *GMDS* deletions associated with tetralogy of Fallot [19,22], the patients in that study were grouped into syndromic vs. non-syndromic. However, it is unclear in which group these two cases with *GMDS* deletions fall. Taken together, there is a degree of phenotypic diversity with regard to the heart phenotype of *GMDS* intragenic deletion carriers, while the extent of extra-cardiac phenotypes is still unclear.

## 5. Conclusions

In summary, we report a patient with Ebstein anomaly with a de novo intragenic *GMDS* deletion. This locus has previously been sporadically associated with Ebstein

anomaly and tetralogy of Fallot. We hypothesize that (large) deletions of this locus lead to loss of function of *GMDS*, which affects fucosylation, and in turn interferes with downstream developmental signaling molecules (e.g., *Notch1*) that are dependent on fucosylation for proper function. Given all the evidence, we conclude that *GMDS* intragenic deletions can be associated with rare cases of congenital heart disease.

**Author Contributions:** The first authors contributed equally: S.M.L.-A.-N. and E.A.V.; Patient's Clinical Care and Follow-up: S.M.L.-A.-N. and E.A.; Conceptualization: S.M.L.-A.-N., E.A.V., M.M.v.H., A.V.P. and J.P.v.T.; Data curation: E.A.V., M.M.v.H., A.V.P., L.T.v.d.V. and J.P.v.T.; Funding Acquisition: not applicable; Project Administration: S.M.L.-A.-N., E.A.V. and M.M.v.H.; Writing: S.M.L.-A.-N., E.A.V., M.M.v.H., A.V.P. and J.P.v.T.; Review and editing: all the authors. All authors have read and agreed to the published version of the manuscript.

**Funding:** This research received no external funding.

**Institutional Review Board Statement:** Under Dutch law, assessment of the study protocol by our ethics committee was not indicated since only genetic and clinical data collected during regular patient care were used. For Dutch research involving CONCOR-data, this is explained on their website https://concor.net/en/aboutconcor/methodology.html (accessed on 1 June 2022).

**Informed Consent Statement:** The parents provided written consent for research use of their data. For Dutch research involving CONCOR-data, this is explained in https://pubmed.ncbi.nlm.nih.gov/16121765/ and on their website https://concor.net/en/aboutconcor/methodology.html (accessed on 1 June 2022).

**Data Availability Statement:** The data presented in this study are available on request from the corresponding author. The data are not publicly available due to privacy reasons.

**Conflicts of Interest:** The authors declare no conflict of interest.

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
