# Peer review of "GMDS Intragenic Deletions Associate with Congenital Heart Disease including Ebstein Anomaly"

_cardiogenetics, doi:10.3390/cardiogenetics13030010_

Round 1
Reviewer 1 Report
In this manuscript, the authors present a 6-yea-old girl with Ebstein anomaly, atrial septum defect, hypoplastic right ventricle and persistent left superior vena cava who has a de novo intragenic ~403 kb deletion of the GMDS gene. However, since after screening 134 more cases with Ebstein anomaly, no additional GMDS genomic aberrations were identified, it is still not clear whether the intragenic deletion of GMDS was the cause of these congenital heart diseases.
Direct evidence of the effect of GMDS on Ebstein anomaly and other congenital heart diseases should be provided to prove these congenital heart diseases were indeed caused by the intragenic deletion of the GMDS.
Author Response
Reviewer 1 Revisited Manuscript ID: cardiogenetics-2259510
Thank you for your time and expertise in reviewing our manuscript ” GMDS Intragenic Deletions Associate with Congenital Heart Disease Including Ebstein Anomaly”, the given comments will definitely improve the quality of the paper.
We thank the reviewer for their comments and expertise.
“In this manuscript, the authors present a 6-year-old girl with Ebstein anomaly, atrial septum defect, hypoplastic right ventricle and persistent left superior vena cava who has a de novo intragenic ~403 kb deletion of the GMDS gene. However, since after screening 134 more cases with Ebstein anomaly, no additional GMDS genomic aberrations were identified, it is still not clear whether the intragenic deletion of GMDS was the cause of these congenital heart diseases”.
With this manuscript we try to shed light on the fact that GMDS might have an important function in cardiac development using a rare cohort of congenital heart disease patients. We are aware that molecular follow-up studies are needed to further pinpoint the mechanism, but we would also like to point out that the current manuscript is an extended case report. Ours is the second case in literature (previous one was published by Sicko et al.), with the same extremely rare, large intragenic and de novo deletion in GMDS and Ebstein anomaly (itself a very rare phenotype). A de novo deletion is in itself already a rare event, let alone two de novo deletions in two independent cases with a very similar rare phenotype.
We argue that this highlights the possibility that this region has something to do with this disease. We had indeed hoped to find more cases with alterations in GMDS, however one should keep in mind that in congenital heart disease, the most frequently found causative gene (NOTCH1) is only present in about 1% of the CHD cases. For Ebstein, a rare subset of CHD, this percentage is even lower. Small numbers of similar genetic findings, like presented here, are therefore what is to be expected for these types of diseases.
Direct evidence of the effect of GMDS on Ebstein anomaly and other congenital heart diseases should be provided to prove these congenital heart diseases were indeed caused by the intragenic deletion of the GMDS.
Given the above, we find it plausible that such a de novo deletion could indeed cause the disease. Moreover, in literature there is good evidence that GMDS is linked to heart development. As discussed, we speculate that such a large deletion (as observed in the two independent cases) would lead to loss of function of GMDS (line 188), and as GMDS is important for fucosylation, which in zebrafish and knock out mice has been linked to a regulatory mechanism (via fucosylation) on NOTCH proteins, in particular NOTCH1 (a crucial heart development gene), we think it is credible that GMDS has a role in normal cardiac development.
In addition to the above de novo cases, there are two CHD patients described in literature by Silverside et al. with smaller deletions in the same region in (line 192). Taken together, we think it is plausible that the GMDS locus is important for normal cardiac development and we agree that further mechanistic studies are needed to prove this point, but these are outside the scope of the current manuscript.
We have now updated the manuscript to reflect this information better.
We are looking forward to your response to our adjustments and hope the manuscript is now ready for publication.

Reviewer 2 Report
Overall the paper is of interest and is very well written. No major changes are needed. I would only request to ensure that the fonts/text size is the same throughout. Also, please specifically declare whether the parents of the child consented to include a photogram of the child with no deidentification. I would suggest that the eyes be covered at least partially (i.e. with a black out line across or blurring of eyes), while retaining the scientific value of showing the photo.
Author Response
Reviewer 2 Revisited Manuscript ID: cardiogenetics-2259510
Thank you for your time and expertise in reviewing our manuscript ” GMDS Intragenic Deletions Associate with Congenital Heart Disease Including Ebstein Anomaly”, the given comments will definitely improve the quality of the paper.
We thank the reviewer for their comments and expertise.
Overall, the paper is of interest and is very well written. No major changes are needed. I would only request to ensure that the fonts/text size is the same throughout. Also, please specifically declare whether the parents of the child consented to include a photogram of the child with no de-identification. I would suggest that the eyes be covered at least partially (i.e., with a black out line across or blurring of eyes), while retaining the scientific value of showing the photo.
We thank reviewer 2 for the comments. We have now adapted the font/text size throughout the manuscript.
The parents have given informed consent to share her medical information, including uncovered photographs. In the initial manuscript the uncovered photos were included because of the clear hypertelorism and mid face hypoplasia (which are likely part of the phenotype). Nonetheless, we have followed your suggestion for privacy and blocked out her eyes, hoping the hypertelorism and mid face hypoplasia are still visible.
We have now updated the manuscript to reflect this information better.
We are looking forward to your response to our adjustments and hope the manuscript is now ready for publication.
Reviewer 3 Report
I read the article and it is really exciting. The regulatory mechanism that GMDS exerts on Notch1 fucosylation undoubtedly needs to be investigated further and could lead to the understanding of the relationship between Ebstein and gestational diabetes. Even if the deletions on chromosome 6 cause Ebstein's disease in a very small group of patients, I agree we could think of performing the CGH array to our patients.Moreover, can authors provide more evidences and a more detailed description of the dysmorphisms evident in the photo? According to me these dysmorphisms can be suspicious for at least three genetic condition: Noonan syndrome, Kabuki (aka Niikawa Kuroki syndrome) and cherubism. Did you perform genetic testing to rule out these conditions?
Another question for the authors: I read in your paper that the prevalence of Ebstein's anomaly is 1 in 20,000 (line 42), while in Mayo Clinic review I found 1 in 200,000 (https://www.ahajournals.org/doi/full/10.1161/circulationaha.106.619338). Both refer to "live births with Ebstein." Can you clarify such difference in epidemiology?
Author Response
Reviewer 3 Revisited Manuscript ID: cardiogenetics-2259510
Thank you for your time and expertise in reviewing our manuscript ” GMDS Intragenic Deletions Associate with Congenital Heart Disease Including Ebstein Anomaly”, the given comments will definitely improve the quality of the paper.
We thank the reviewer for their comments and expertise.
- “I read the article and it is really exciting. The regulatory mechanism that GMDS exerts on Notch1 fucosylation undoubtedly needs to be investigated further and could lead to the understanding of the relationship between Ebstein and gestational diabetes. Even if the deletions on chromosome 6 cause Ebstein's disease in a very small group of patients, I agree we could think of performing the CGH array to our patients. “
- The regulatory mechanism of GMDS on heart development is likely complex and indeed needs to be further investigated. We, and others, will hopefully show more relationships with other (congenital) diseases in the future.
- Indeed, it is worthwhile to perform an ArrayCGH or similar techniques on all Ebstein’s patients, as in our case (a small Caribbean island), 1 out of 6 had this deletion.
- “Moreover, can authors provide more evidences and a more detailed description of the dysmorphisms evident in the photo? “
- Thank you for this remark, indeed because of the dysmorphisms shown on figure 1A clinical geneticist (Co-author MVH) did a thorough physical exam. The dysmorphic features in our patient consisted of: Hypertelorism, long flat philtrum and mid face hypoplasia and divergent eyes, a difference in size of the globe, besides an anterior placed anus. No other dysmorphisms were observed.
- The parents have given informed consent for sharing her information, including uncovered photographs because of the clear hypertelorism and mid face hypoplasia. However, as also pointed out by the other reviewer, we have adopted now blocked out her eyes, hoping the hypertelorism and mid face hypoplasia are still visible.
- According to me these dysmorphisms can be suspicious for at least three genetic condition: Noonan syndrome, Kabuki (aka Niikawa Kuroki syndrome) and cherubism. Did you perform genetic testing to rule out these conditions?
The reviewer raises an important point. We had Noonan Syndrome in our differential diagnosis and performed additional testing. A next-generation sequencing (NGS) targeted structural heart disease gene panel was performed, which consisted of 46 genes, including Noonan-syndrome related genes (BRAF, KRAS, LZTR1, MAP2K1, NRAS, PTPN11, RAF1, RASA2, RIT1, SOS1, SOS2). There were no abnormalities identified.
- We have not tested our patient for Kabuki syndrome since facial features were not suggestive for this syndrome and this condition is more associated with left sided heart defects whereas Ebstein’s anomaly is primarily a right heart disease.
- No abnormal bone formations of the mandibula jaws, other bones or signs of inflammatory features have been noted in our patient clinically or on radiology. There is no clear relationship with congenital heart disease in the literature, except for cases related with Noonan syndrome. We have not tested her for cherubism as she is now above 7 years of age and still shows no abnormal bone formations.
Another question for the authors:
I read in your paper that the prevalence of Ebstein's anomaly is 1 in 20,000 (line 42), while in Mayo Clinic review, I found 1 in 200,000 (https://www.ahajournals.org/doi/full/10.1161/circulationaha.106. 619338). Both refer to "live births with Ebstein." Can you clarify such difference in epidemiology?
- “Indeed in the original literature, the reported live birth prevalence of Ebstein Anomaly is 1:200.000 (Attenhofer Jost)(line 44) , but more recent population studies have reported a lower birth prevalence of 1: 20.000.(Correa-Villaseñor, Lupo, Pradat(2-4) )(line 45). This has been attributed to different factors, amongst others better technology and earlier (even prenatal) and better diagnoses (Boyle).
- Also see differences in ascertainment methods and classifications among different institutes and countries which also in recent periods can give a different outcome. For instant only diagnosis below the age of 1 year with echocardiography, catheterization, surgery and/or pathology. This shift in birth prevalence, has been mentioned in the introduction of the paper by Sicko and Boyle.(6,5) (line: 45-46)
We have now updated the manuscript to reflect this information better.
We are looking forward to your response to our adjustments and hope the manuscript is now ready for publication.
Round 2
Reviewer 1 Report
The authors have addressed my concern.